# Assessment of Land Use and Land Cover Changes on Soil Erosion Using Remote Sensing, GIS and RUSLE Model: A Case Study of Battambang Province, Cambodia

Taingaun Sourn [1,2,*], Sophak Pok [2], Phanith Chou [3], Nareth Nut [1,4], Dyna Theng [4] and P. V. Vara Prasad [5]

[1]   Graduate School, Royal University of Agriculture, Phnom Penh 12400, Cambodia; nnareth@rua.edu.kh
[2]   Faculty of Land Management and Land Administration, Royal University of Agriculture, Phnom Penh 12400, Cambodia; poksophak@gmail.com
[3]   Faculty of Development Studies, Royal University of Phnom Penh, Phnom Penh 12150, Cambodia; chou.phanith@rupp.edu.kh
[4]   Faculty of Agricultural Engineering, Royal University of Agriculture, Phnom Penh 12400, Cambodia; thdyna@rua.edu.kh
[5]   Department of Agronomy and Sustainable Intensification Innovation Lab (SIIL), Kansas State University, Manhattan, KS 66506, USA; vara@ksu.edu
*   Correspondence: sourntaingauns@gmail.com; Tel.: +855-15-455-686

**Abstract:** Soil erosion causes land degradation which negatively impacts not only natural resources but also livelihoods of people due to low agricultural productivity. Cambodia is prone to soil erosion due to poor agricultural practices. In this research we use Battambang province as a case study to quantify impact of land use and land cover change (LULC) on soil erosion. This study assessed the impact from LULC changes to soil erosion. LULC change maps were analyzed based on Landsat satellite imagery of 1998, 2008, and 2018, computed in QGIS 6.2.9, while the soil erosion loss was estimated by the integration of remote sensing, GIS tools, and Revised Universal Soil Loss Equation (RUSLE) model. The results showed that the area of agricultural land of Battambang province significantly increased from 44.50% in 1998 to 61.11% in 2008 and 68.40% in 2018. The forest cover significantly decreased from 29.82% in 1998 to 6.18% in 2018. Various soil erosion factors were estimated using LULC and slope. Based on that, the mean soil loss was 2.92 t/ha.yr in 1998, 4.20 t/ha.yr in 2008, and 4.98 t/ha.yr in 2018. Whereas the total annual soil loss was 3.49 million tons in 1998, 5.03 million tons in 2008, and 5.93 million tons in 2018. The annual soil loss at the agricultural land dramatically increased from 190,9347.9 tons (54%) in 1998 to 3,543,659 tons (70.43%) in 2008 and to 4,267,439 tons (71.91%) in 2018 due to agricultural land expansion and agricultural practices. These losses were directly correlated with LULC, especially agricultural land expansion and forest cover decline. Our results highlight the need to develop appropriate land use and crop management practices to decrease land degradation and soil erosion. These data are useful to bring about public awareness of land degradation and alert local citizens, researchers, policy makers, and actors towards land rehabilitation to bring the area of land back to a state which is safe for increasing biodiversity and agricultural productivity. Measures to reduce or prevent soil erosion and the use of conservation agriculture practices, along with water and soil conservation, management, agroforestry practices, vegetation cover restoration, the creation of slope terraces, and the use of direct sowing mulch-based cropping systems should be considered.

**Keywords:** soil erosion; land use and land cover change; RUSLE; GIS; land degradation; agricultural land; forestland; Cambodia

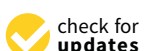



## 1. Introduction

Soil erosion due to land degradation negatively impacts not only natural resources but also livelihoods of people as a result of low agricultural productivity [1–5]. Land degradation reduces economic yield from agricultural, forest, grass, or shrub lands, decreases soil

fertility, biodiversity and ecosystem services, and ultimately farmers' economy [6–8]. The major factors of land degradation are soil erosion, loss of topsoil with fertility, decrease of agricultural production, poor water quality, and flooding [6,9,10]. Furthermore, developing countries are currently faced with immediate concerns that relate to land degradation, freshwater shortages, food insecurity, and air and water pollution. Climate change will further exacerbate these concerns, leading to rapid water shortages, land degradation, and desertification [11].

Generally, the negative impact of soil erosion has two kinds, including the on-site or long-term adverse impacts on soil quality and productivity, and the offsite or short-term impacts [12–14]. Soil erosion impacts the decrease in agricultural productivity through a decrease in soil structure and reduction in effective rooting depth, loss of plant nutrients and soil organic carbon (SOC), loss of plant-available water and available water content (AWC), loss of land area, and seedling damage, resulting in serious problems such as drought stress, crust formation, compaction, and poor emergence of seedling and crop stands [12,15]. Lal [16] reported that USD 40 million was annually expended for extra fertilizer cost to restore soil quality impacted by erosion in order to increase agricultural yields in the United States. Similarly, it was reported that in southern Alberta, crop yield decreased from 10% to 39%, while 5–20 cm top soil was eroded [17]. Furthermore, soil erosion also impacts water quality [2,12]. Moreover, the intensive utilization of agrochemicals is an inevitable effect of increasing population pressures in Asia and Africa. The risks of the adverse consequence of erosion-induced transport of chemicals into water bodies are likely to increase [12].

In the Cambodian context, forest clearance, LULC change, agricultural land expansion, and improper crop, soil, and water management practices resulted in land degradation [10]. The northwestern part of Cambodia is facing severe land degradation and soil erosion [18]. According to Chuenchum et al. [2], FAO [19], and Van Oost et al. [20], over 20 t/ha/yr topsoil was eroded from agricultural land in many tropical regions and countries in Southeast Asia. For example, soil erosion of 6.5 t/ha.yr was reported in Thailand [21] and 2 t/ha.yr in Viet Nam in 1992 [22]. Chuenchum et al. [2] estimated that annual soil erosion along the Mekong River ranged from 7 to 100 t/ha.yr in 2019. Cambodia shares similarities with many countries in the region (e.g., Thailand, Viet Nam, and Laos) and is impacted by soil erosion [2,23,24]. Previous findings showed that the average annual soil erosion rate in Cambodia was in the range from 7 to less than 100 t/ha.yr [2]. The problem is more serious upland due to steep topography, rapid deforestation, and agricultural expansion [18,24–26]. For example, a cassava field eroded around 60 to 119 t/ha/262 days in the upland area of Battambang [27]. Another study estimated that in land with more than 30-degree slope, the soil erosion ranged from 16.3 t/ha.yr in 2002 to 27.6 t/ha.yr in 2015 due to LULC change [25].

The spatial distribution and mapping of soil erosion in steep slopes is important information needed to develop a sustainable plan for appropriate land management, agricultural management focused on minimizing land degradation [28]. There are several models and tools such as the Agricultural Policy/Environmental eXtender (APEX), Soil and Water Assessment Tool (SWAT), Universal Soil Loss Equation (USLE) or Revised Universal Soil Loss Equation (RUSLE), Environmental Policy Integrated Climate (EPIC), Water Erosion Prediction Project (WEPP), the Erosion Productivity Impact Calculator (EPIC), and the Agricultural Nonpoint Source (AGNPS) to estimate soil loss [25,28–30]. Neges et al. [28] concluded that a combination of remote sensing and geographic information system (GIS) with other models is beneficial. For example, RUSLE is commonly used to study soil erosion in different regions around the world. The RUSLE with the integration of GIS was developed by Wischmeier and Smith [31]. It was employed by many researchers in the Mekong regions [2,23,25,32], Thailand [21,33], Ethiopia [9,28,34–37], China [38–40], and Nepal [41] to predict soil erosion. The RUSLE was employed to predict soil erosion in regions when the measured data are limited [28,30,31,42]. There are five factors of the RUSLE model, namely rainfall erosivity (R), soil erodibility (K), topography (LS), cover and

management (C), and support practice (P) used to estimate the long-term average erosion rate, annually [31].

Many researchers [2,23,30] used the combination of GIS and remote sensing with the RUSLE model to estimate soil erosion in Mekong regions. However, these studies were not conducted over long time periods, using cover and management factor (C) estimated using NDVI based on the satellite images. Nut et al. [25] estimated soil erosion and mapped soil risks in Stung Sangkae catchment, Cambodia. However, the cover and management factor (C) and supported practice (P) were extracted from existing LULC, made by Japan International Cooperation Agency (JICA) in 2002 and Mekong River Commission (MRC) in 2015 during the estimation of soil loss using RUSLE.

In our recent study [26], we evaluated LULC changes and its drivers in Battambang province from 1998 to 2018. However, we did not quantify the impact of LULC on soil erosion. A complete picture of assessing the impact of LULC on soil erosion for a long-term period in the entire Battambang province is not well understood and needs attention. It is important to thoroughly understand the estimates and pattern of soil erosion risk for developing better agricultural management practices, improved land use policies, and efficient management of natural resources in order to rehabilitate the land. In this regard, this research study for assessing the impact of LULC on soil erosion in the year 1998, 2008, and 2018 was carried out in the entire Battambang province of Cambodia using a combination of GIS, remote sensing data, and RUSLE model. The study attempted to answer three specific questions: (1) How did the spatial distribution pattern of soil erosion change from 1998 to 2018? (2) What was the long-term annual average soil loss rate? (3) Where are the prioritized zones of soil erosion risk for planning and implementation of conservation measures? To answer these questions, we used Landsat 5 TM and Landsat 8 OLI images to produce LULC maps. These were used to estimate C factor, and the combination between LULC and slope was used to predict P factor.

## 2. Materials and Methods

### 2.1. Description of Study Area

The study area of Battambang province is located in the northwestern part of Cambodia (Figure 1a). It was categorized by four ecological zones: upland area, semi-upland area, lowland area, and floodplain along Tonle Sap Lake and was noted to have a mixture of various land uses. According to JICA 2002 [43], there were seven classes of LULC, namely: (1) built-up area, (2) water feature, (3) grassland, (4) shrubland, (5) agricultural land, (6) barren land, and (7) forest cover. In addition, currently, it also has economic land concession (ELC) and protected areas (Figure 1b). In only one decade (2006 to 2016), 65% of forest cover was lost, while agricultural land dramatically rose from 1% in 1997 to 61% in 2016 in the northwestern uplands of Battambang and Pailin provinces [24]. The deforestation of the uplands negatively impacted soil erosion. The study area has a major Sangkae stream with a catchment area of 605,170 hectares [25]. It also has approximately 3951 streams/canals with a total 7313 km length, and the maximum and mean length of the canal were 47 km and 1.90 km, respectively (Figure 1a).

The elevation in the study area ranged from 0 to 1333 m above mean sea level (MSL) [26,44]. The average annual rainfall was 1280 mm between 1995 and 2018 (Figure 2). The maximum yearly rainfall was 1566 mm in 2011 and minimum yearly rainfall was 947 mm in 2014 [45]. Battambang, not different from the national climate conditions, is under the influence of tropical monsoon climate, which consists of two main seasons: rainy season and dry season [46]. The rainy season starts in May and ends in October, while the dry season period is from November to April [46]. The province has an average annual temperature of 27.7 °C [45].

More than 50% of the total area had a very gentle slope to a gentle slope, and approximately 22% of total areas ranged from strong slope to very steep slope (Table 1).

A significant increase in population was noticed for the province, from 793,129 in 1998 to 997,169 in 2018 [47], which can negatively impact the land use, natural resources, environment, and particularly, soil erosion. There are nine different soil types: Acrisols,

Arenosols, Cambisols, Ferralsols, Fluvisols, Gleysols, Lixisols, Luvisols, and Vertisols. However, approximately 94% of the total area was covered predominantly by Acrisols, Fluvisols, and Cambisols.

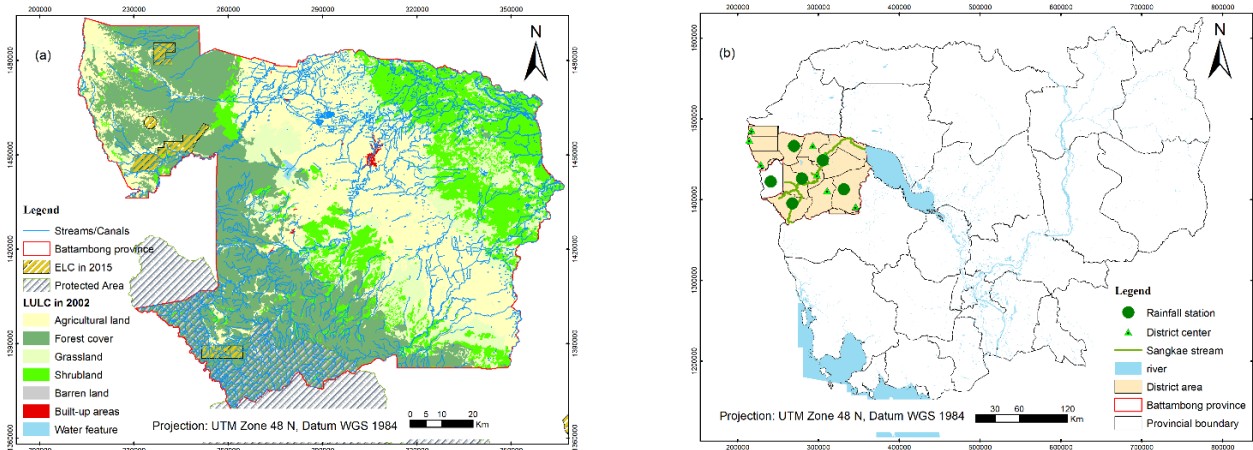

**Figure 1.** Study area: (**a**) different land use and land cover (LULC) areas in 2002, protected areas, economic land concession (ELC), and streams/canals; and (**b**) Battambang Province in Cambodia and rainfall stations.

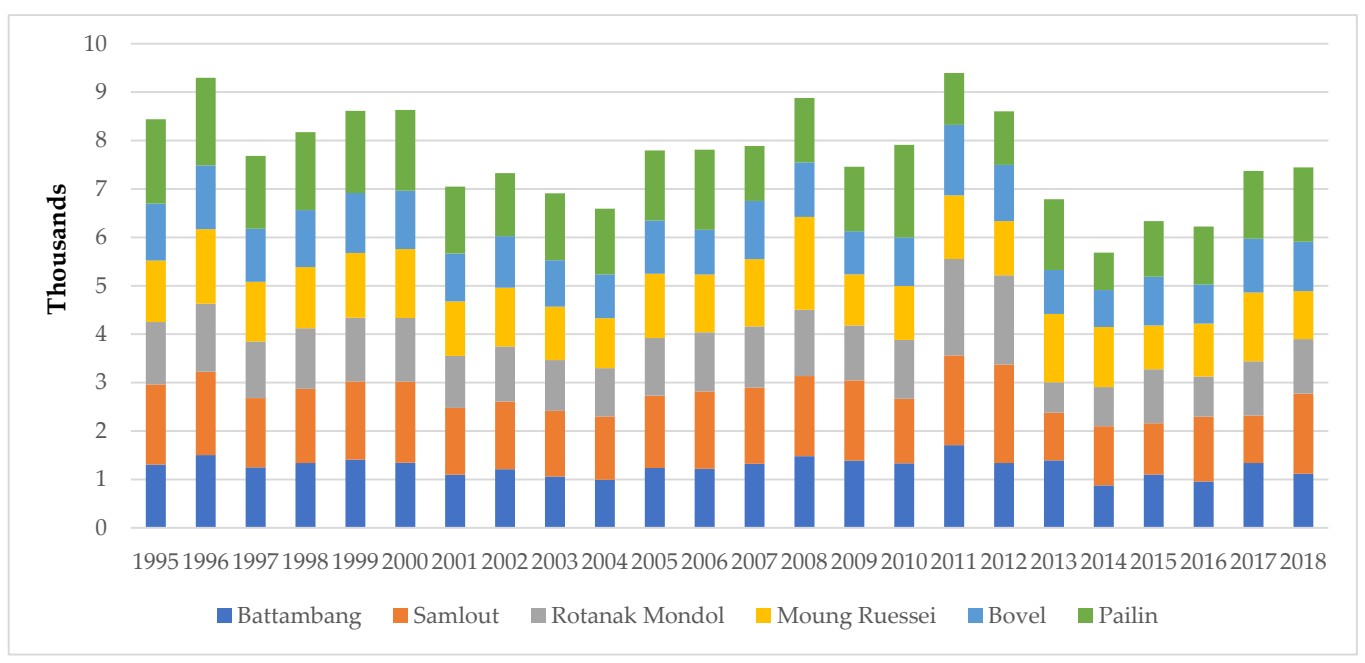

**Figure 2.** Rainfall from 1995–2018 extracted from six weather stations combined with satellite image.

## 2.2. Land Use/Land Cover

The main data for LULC change categories in the study area were obtained from Landsat images. These included Landsat 5 TM and Landsat 8 OLI scenes for the years 1998, 2008, and 2018. The Landsat images were derived from the United States Geological Survey (USGS) website [48]. All Landsat data were acquired in the same dry season from December to April. Accuracy assessment was conducted using a total of 121, 163, and 317 validation points randomly selected in 1998, 2008, and 2018, respectively. This approach was followed by many researchers and is well documented [49–52]. The reference data for each LULC class in 2018 were collected by field visit, using drone and handheld GPS.

However, the reference data for 1998 and 2008 were obtained from the existing maps of land use of 1993 from the Geographic Department and the land use map of 2002 from the JICA. The forest cover maps of 2002, 2006, and 2010 were obtained from Ministry of Agriculture, Forestry and Fisheries (MAFF) and Google Earth images supplemented by field visits, focused group discussions, key informant surveys, and in-depth interviews with elders in the study area. Overall accuracy, user's accuracy, producer's accuracy, and Kappa coefficient were defined as the common measures of classification accuracy obtained from the error matrix [50,53,54].

**Table 1.** Slope severity in the study area based on FAO slope classification.

| Slope Severity | Slope Class (%) | Area | |
| --- | --- | --- | --- |
| | | (ha) | (%) |
| Flat to very gentle slope | 0–2 | 326,342 | 27.11 |
| Gentle slope | >2–5 | 314,096 | 26.1 |
| Medium slope | >5–10 | 302,890 | 25.16 |
| Strong slope | >10–15 | 121,368 | 10.08 |
| Moderately steep | >15–30 | 93,715 | 7.79 |
| Steep | >30–60 | 39,972 | 3.32 |
| Very steep | >60 | 5245 | 0.44 |
| Total | | 1,203,628 | 100 |

*2.3. Soil Erosion Estimation*

The soil erosion estimation model of RUSLE [31] was integrated with spatial analysis of GIS and remote sensing [55,56]. According to Bahadur [57]; Chuenchum et al. [2,32]; Thuy and Lee [23], the model was used effectively and it was simple to estimate soil erosion in the Mekong River region. The model is based on five parameters and its equation is shown below (Equation (1)).

$$A = R \times K \times LS \times C \times P \tag{1}$$

where: A is the soil loss in t/ha/yr, R is the rainfall erosivity factor in (MJ mm/ha/hr/year); K is the soil erodibility factor (t ha hr/ha/(MJ mm)); LS is the topographic factor (dimensionless); C is the cropping management factor (dimensionless); and P is the soil and water conservation practices factor (dimensionless). The soil erosion estimations were analyzed by ArcGIS 10.3. The conceptual framework is given in Figure 3.

2.3.1. Rainfall Erosivity (R) Factor Prediction

Rainfall data (1995–2018) were obtained from Ministry of Water Resources and Meteorology (MoWRM) and downloaded from satellite image by conducting bias correction to estimate the rainfall erosivity (R) factor of RUSLE based on the formula established by [58] and used by [59]. It was used to compute in ArcGIS 10.3 raster calculation.

$$R = 38.5 + 0.35 \, r \tag{2}$$

where r is annual rainfall (mm)

2.3.2. Soil Erodibility (K) Factor Prediction

The *K* factor was estimated based on soil characteristics and soil properties; in particular, soil particle size and soil texture ranged from 0 to 1. The local existing soil data in Cambodia, and particularly Battambang province, are rarely available. Thus, the digital soil map was obtained from the Soil Grids database of ISRIC-World Soil Information [60]. The spatial resolution of the soil type map was 250 m. This soil map data was used by researchers in the world for areas such as Mekong region and Ethiopia. Ref. [61] developed the equation to estimate *K* factor, which was used in this study.

$$K_{Rusle} = f_{csand} \times f_{orgC} \times f_{hisand} \tag{3}$$

$$f_{csand} = 0.2 + 0.3 \, Exp\left[-0.256 \times Ms \times \left(1 - \frac{M_{silt}}{100}\right)\right] \tag{4}$$

$$f_{cl-si} = \left(\frac{M_{silt}}{Mc + M_{silt}}\right)^{0.3} \tag{5}$$

$$f_{orgC} = \frac{0.0256 \times Mo}{Mo + Exp[3.72 - (2.95 \times Mo)]} \tag{6}$$

$$fhisand = 1 - 0.7 \times \frac{1 - \frac{Ms}{100}}{\left(1 - \frac{Ms}{100}\right) + \exp\left[-5.51 + 22.9 \times \left(1 - \frac{Ms}{100}\right)\right]} \tag{7}$$

where $K_{Rusle}$ is the soil erodibility factor, $f_{csand}$ is a function of the high coarse sand content of the soil, $f_{orgC}$ is a function of the organic carbon content of the soil, $f_{hisand}$ is the function of high sand content in the soil, $Ms$ is the % sand, $M_{silt}$ is % silt, $f_{cl-si}$ is a function of the clay and silt of the soil, $Mc$ is the % clay, and $Mo$ is % organic matter.

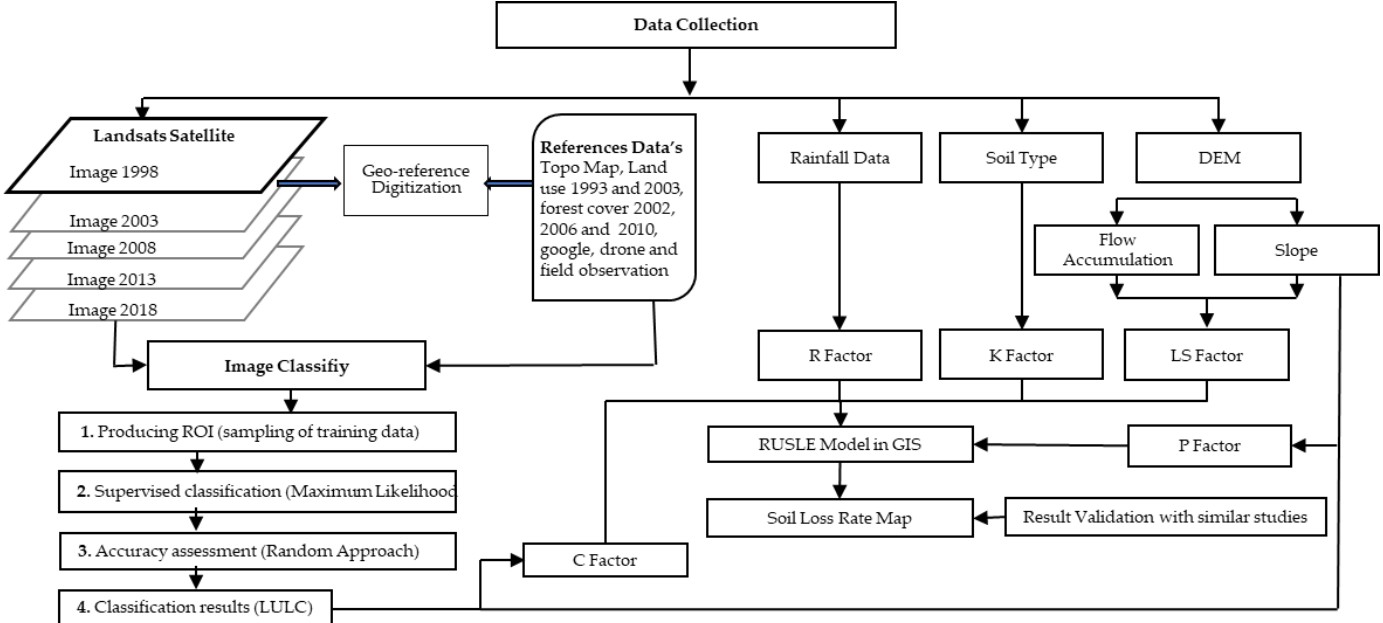

**Figure 3.** Flowchart of LULC change and soil erosion analysis in Battambang province.

### 2.3.3. Topographic Parameters (LS) Factor Prediction

The two significant parameters of topographic factors, slope length and steepness, were determined by GIS application with 30 m resolution using Digital Elevation Model (DEM) [62] derived from USGS. The LS factor was estimated as per equation (Equation (8)) [63].

$$LS = (\text{Flow Accumulation} \times 30/22.1)^m) \times (0.065 + 0.045 \times \text{slope} + 0.0065 \times \text{slope} \times \text{slope}) \tag{8}$$

where m values, of the slope classes: <1, $1 \leq$ Slope < 3, $3 \leq$ Slope < 5 and > 5 are 0.2, 0.3, 0.4, and 0.5, respectively.

### 2.3.4. Land Cover and Management (C) Factor Prediction

Land cover and management (C) factor significantly impacts soil erosion [34]. The C factor in this study was calculated from LULC maps from 1998 to 2018 and existing data from other researchers [34,35,64]. Based on LULC classes, the C factor values of agricultural land, forest land, shrubland, grassland, built-up area, and barren land classes are 0.50, 0.01, 0.014, 0.08, 0.1, and 0.35, respectively (Table 2).



**Table 2.** Land cover and management (C) factor for different land use and land cover (LULC) types.

| LULC Type | C Factor | References |
|---|---|---|
| 1. Built-up area | 0.1 | [2,25,64] |
| 2. Water feature | 0.01 | [2,25,64] |
| 3. Grassland | 0.08 | [25,64] |
| 4. Shrubland | 0.014 | [25,64] |
| 5. Agricultural land | 0.5 | [25,64] |
| 6. Barren land | 0.35 | [2,25,64] |
| 7. Forest cover | 0.01 | [2,25,64] |

2.3.5. Management Practices (P) Factor Prediction

The management practices (P) factor is used to reduce the impact of LULC on soil erosion by managing water flow through agricultural conservation practices which includes contouring, buffer strips, and terraces contour farming [5,31,36,65]. Two methods commonly used to estimate P factors include the following: the first approach is the agricultural conservation practices and the second approach is the combination between slope and LULC [66]. In this study, the second approach was used to determine P factor of RUSLE. The annual soil loss rate is affected by the P factor with soil and water conservation [34,65]. The range of the P factor is from 0 to 1, which is the highest value. Referring to [34], adopted RUSLE P factor values for watershed conservation practices are given in Table 3. In addition, a combinatorial and spatial analyst tool of ArcGIS 10.3 was used to overlay slope and LULC types to estimate P factors, and then a look-up spatial analyst tool was used to extract the map layer of the P values.

**Table 3.** The management practice (P) factor of Revised Universal Soil Loss Equation (RUSLE) used in this study [31].

| Land Use and Land Cover Categories | Slope % | P factor |
|---|---|---|
| Agricultural land | 0 to 5 | 0.1 |
| | >5 to 10 | 0.12 |
| | >10 to 20 | 0.14 |
| | >20 to 30 | 0.19 |
| | >30 to 50 | 0.25 |
| | >50 to 100 | 0.33 |
| Other land use and land cover categories | All | 1.0 |

## 3. Results

### 3.1. Land Use and Land Cover Change Detection

The LULC data and maps were categorized into seven classes, namely, built-up area, grassland, shrubland, agricultural land, barren land, forest cover, and water feature (Table 4; Figure 4). LULC maps of 1998, 2008, and 2018 were made from multi-temporal Landsat images with significant overall accuracy and Kappa coefficients [67,68]. They were 93% and 0.92 in 1998, 94% and 0.93 in 2008, and 84% and 0.80 in 2018, respectively. Detailed and more granular changes are shown for five different times series (1998, 2003, 2008, 2013, and 2018) published before [26]. Here, we are providing a summary of area and percentage change (Table 4) and spatial distribution (Figure 4) of major LULC changes between the three time frames (1998, 2008, and 2018)

The LULC category map of 1998 shows that the second largest area occupied in Battambang was forest cover with 359,000 ha (30%), followed by agricultural land with 535,600 ha (45%), while the smallest area was barren land with 16 ha (0.004%). In 2008, forest cover decreased, while there was rapid expansion of agricultural land (61.11%) and other classes, namely, grassland (17%) and shrubland (15%). There were also increases in water feature

(0.4%), built-up area (0.03%), and barren land (0.02%). In the year 2018, there was further expansion in agricultural land and shrubland to 68% and 18%, respectively. However, the extent of forest cover and grassland experienced a decrease to approximately 6% (Table 4).

**Table 4.** Total area and percentage of LULC by the years 1998, 2008, and 2018 and LULC changes during 2008–1998, 2018–2008, and 2018–1998.

| LULC Type | Area (1998) | | Area (2008) | | Area (2018) | | 1998–2008 | | 2008–2018 | | 1998–2018 | |
| --- | --- | --- | --- | --- | --- | --- | --- | --- | --- | --- | --- | --- |
| | Hectares | % | Hectares | % | Hectares | % | Hectares | % | Hectares | % | Hectares | % |
| 1. Built-up area | 48 | 0.00 | 302 | 0.03 | 4698 | 0.39 | 300 | 527 | 4400 | 1455 | 4600 | 9651 |
| 2. Water feature | 2309 | 0.19 | 4707 | 0.39 | 10,637 | 0.88 | 2400 | 104 | 5900 | 126 | 8300 | 361 |
| 3. Grassland | 151,753 | 12.61 | 208,051 | 17.29 | 75,683 | 6.29 | 56,300 | 37 | −132,400 | −64 | −76,100 | −50 |
| 4. Shrubland | 154,916 | 12.87 | 178,717 | 14.85 | 213,575 | 17.74 | 23,800 | 15 | 34,900 | 20 | 58,700 | 38 |
| 5. Agricultural land | 535,627 | 44.50 | 735,584 | 61.11 | 823,225 | 68.40 | 200,000 | 37 | 87,600 | 12 | 287,600 | 54 |
| 6. Barren land | 16 | 0.00 | 224 | 0.02 | 1395 | 0.12 | 200 | 1327 | 1200 | 521 | 1400 | 8766 |
| 7. Forest cover | 358,960 | 29.82 | 76,042 | 6.32 | 74,416 | 6.18 | −282,900 | −79 | −1600 | −2 | −284,500 | −79 |
| Grand Total | 1,203,628 | | 1,203,628 | | 1,203,628 | | | | 1,203,628 | | | |

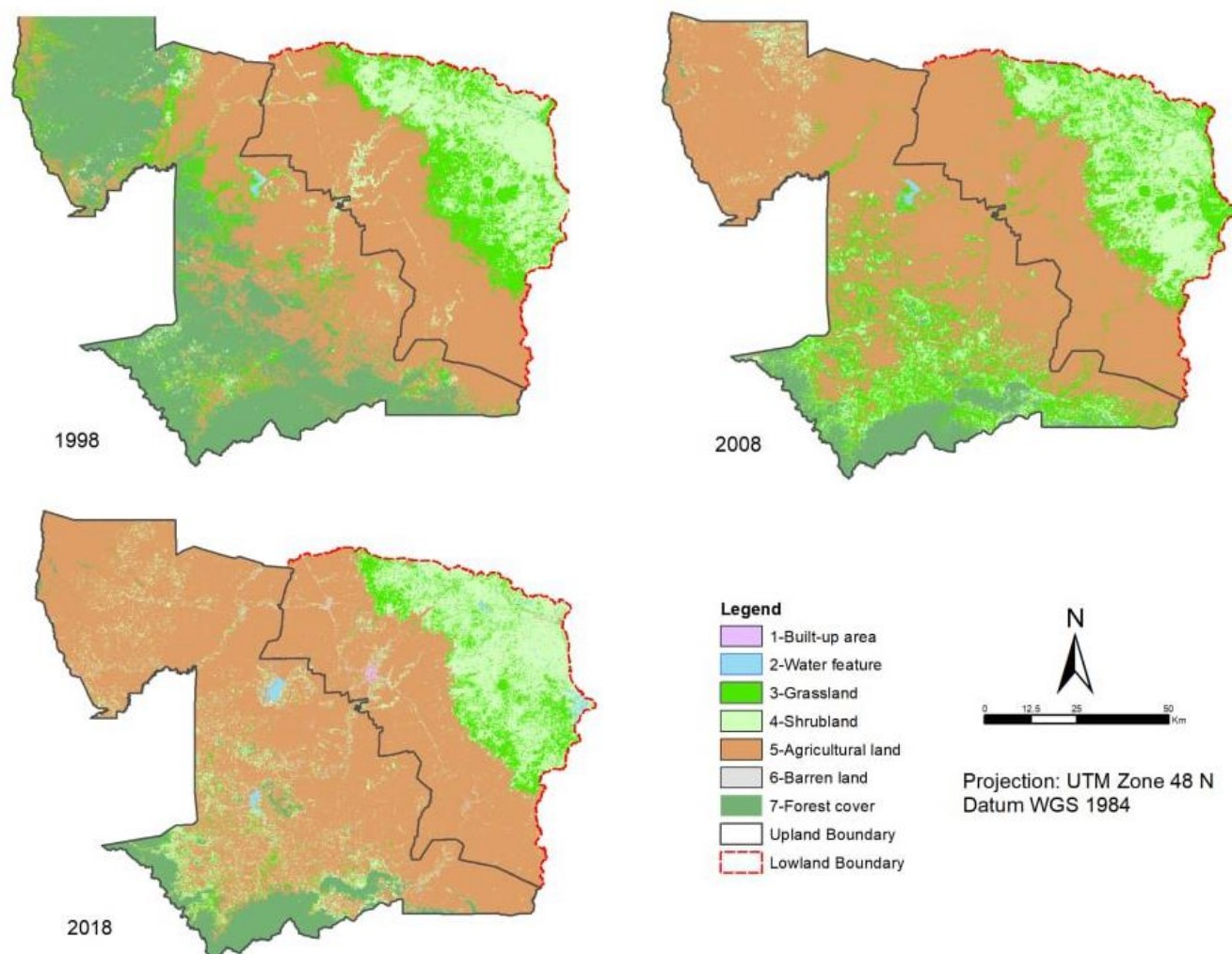

**Figure 4.** Spatial distribution patterns of LULC of the Battambang province in 1998, 2008, and 2018.

*3.2. Soil Erosion Loss Estimation*

3.2.1. Rainfall Erosivity (R) Factor

The average annual rainfall in 1995–2018 (Table 5) was obtained from six stations of the Ministry of Water Resources and Meteorology (MoWRM) of Cambodia and the satellite

image was obtained by performing bias correction to be used to estimate erosivity R factor for the study area based on Equation (2). The interpolation tool (Inverse Distance Weighting (IDW)) of ArcGIS 10.3 was used to compute the R factor with a spatial resolution of 30 m. Figure 5a illustrates the range of R factor value from 412.17 to 556.28 MJ.mm/(ha.hr.yr) with an average of 480.73 MJ.mm/(ha.hr.yr) and the standard deviation of 26.26. The highest R factor was recorded at the upland of the southwestern study area, while the lowest R factor was recorded at some parts of the northern study area in low land (Table 5).

**Table 5.** Meteorological station, average of annual rainfall and R factor.

| Station Name | Location | | Province | Elevation (m) | Average of Annual Rainfall (1995–2018) | R Factor (MJ/(mm.ha.hr.yr) |
| | Longitude | Latitude | | | | |
| --- | --- | --- | --- | --- | --- | --- |
| Battambang | 103.204 | 13.0989 | Battambang | 94 | 1263.77 | 480.82 |
| Samlout | 102.8594 | 12.61453 | Battambang | 153 | 1479.38 | 556.28 |
| Rotanak Mondol | 102.9674 | 12.89267 | Battambang | 258 | 1203.05 | 459.57 |
| Moung Ruessei | 103.4457 | 12.77753 | Battambang | 29 | 1251.06 | 476.37 |
| Bovel | 102.875 | 13.25614 | Battambang | 30 | 1067.62 | 412.17 |
| Pailin | 102.6115 | 12.85589 | Pailin | 95 | 1414.50 | 533.58 |

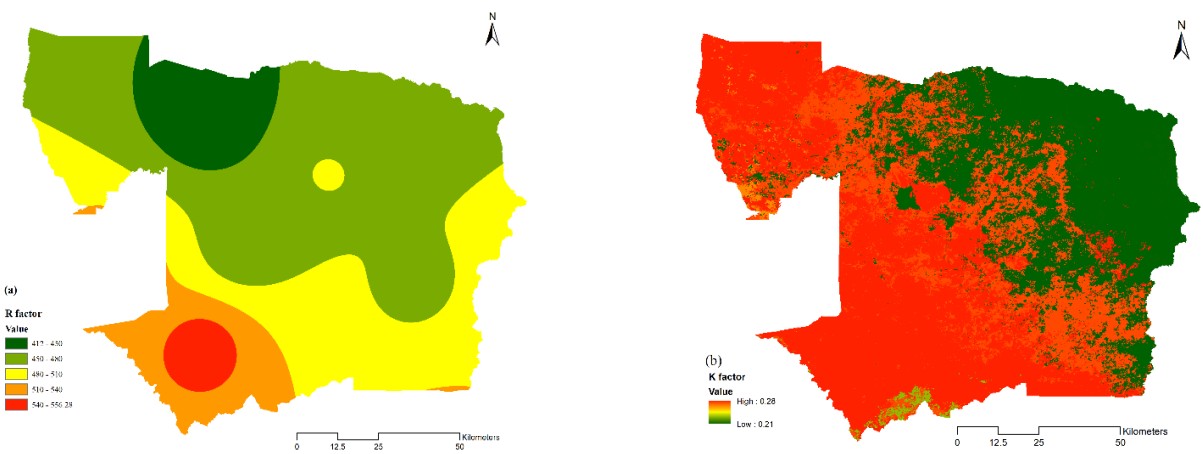

**Figure 5.** Spatial map of (**a**) rainfall erosivity factor (R) and (**b**) soil erodibility factor (K).

### 3.2.2. Soil Erodibility (K) Factor

There are nine soil types: Acrisols, Arenosols, Cambisols, Ferralsols, Fluvisols, Gleysols, Lixisols, Luvisols, and Vertisols (Table 6). The soil type of Acrisols (Loam) was the largest soil area with 42.1% (507,041 ha), followed by Fluvisols (Clay Loam) with 30.7% (369,122 ha). In contrast to the most prominent site of soil types, the size of Arenosols and Lixisols was less than 500 hundred hectares. The soil erodibility (K) factor value of the study area ranged from 0.21 to 0.28 t/(hr.MJ.mm) (Figure 5b).

### 3.2.3. Topographic (LS) Factor

The flowing water from precipitation and runoff made soil erosion heavier depending on topographical factor [2]. Equation (8) was used to estimate the LS factor by using 30 m spatial resolution DEM, acquired from the Advanced Spaceborne Thermal Emission and Reflection (ASTER) global digital elevation map via the website of USGS [44]. The altitude range in Battambang province is from 0 to 1341 m above mean sea level (MSL). The high elevation site was mainly located in the mountain at the southwest of the province. The result of the LS factor estimation ranged from 0 to 321.22 (Figure 6). Its average and standard deviation were 0.53 and 1.97, respectively. In addition, the range of LS factor from 0 to 5 occupied predominantly 98.22% (1,182,154 ha), while 50–312.22 of LS was almost zero percentage (Table 7). Furthermore, the highest LS factor was observed at the mountain and the lowest LS factor was mostly located in the lowland.

**Table 6.** Soil types, soil texture, and K factors.

| No. | Soil Type | Soil Texture | Area | | K Factor |
| --- | --- | --- | --- | --- | --- |
| | | | (ha) | % | t/(hr.MJ.mm) |
| 1 | Acrisols | Loam | 507,041 | 42.1% | 0.27 |
| 2 | Arenosols | Loam | 21 | 0.0% | 0.27 |
| 3 | Cambisols | Clay Loam | 262,484 | 21.8% | 0.27 |
| 4 | Ferralsols | Sandy Clay Loam | 7227 | 0.6% | 0.23 |
| 5 | Fluvisols | Clay Loam | 369,122 | 30.7% | 0.21 |
| 6 | Gleysols | Clay Loam | 42,568 | 3.5% | 0.22 |
| 7 | Lixisols | Loam | 31 | 0.0% | 0.28 |
| 8 | Luvisols | Clay Loam | 6814 | 0.6% | 0.26 |
| 9 | Vertisols | Clay Loam | 803 | 0.7% | 0.26 |
| | Total | | 1,203,628 | 100% | |

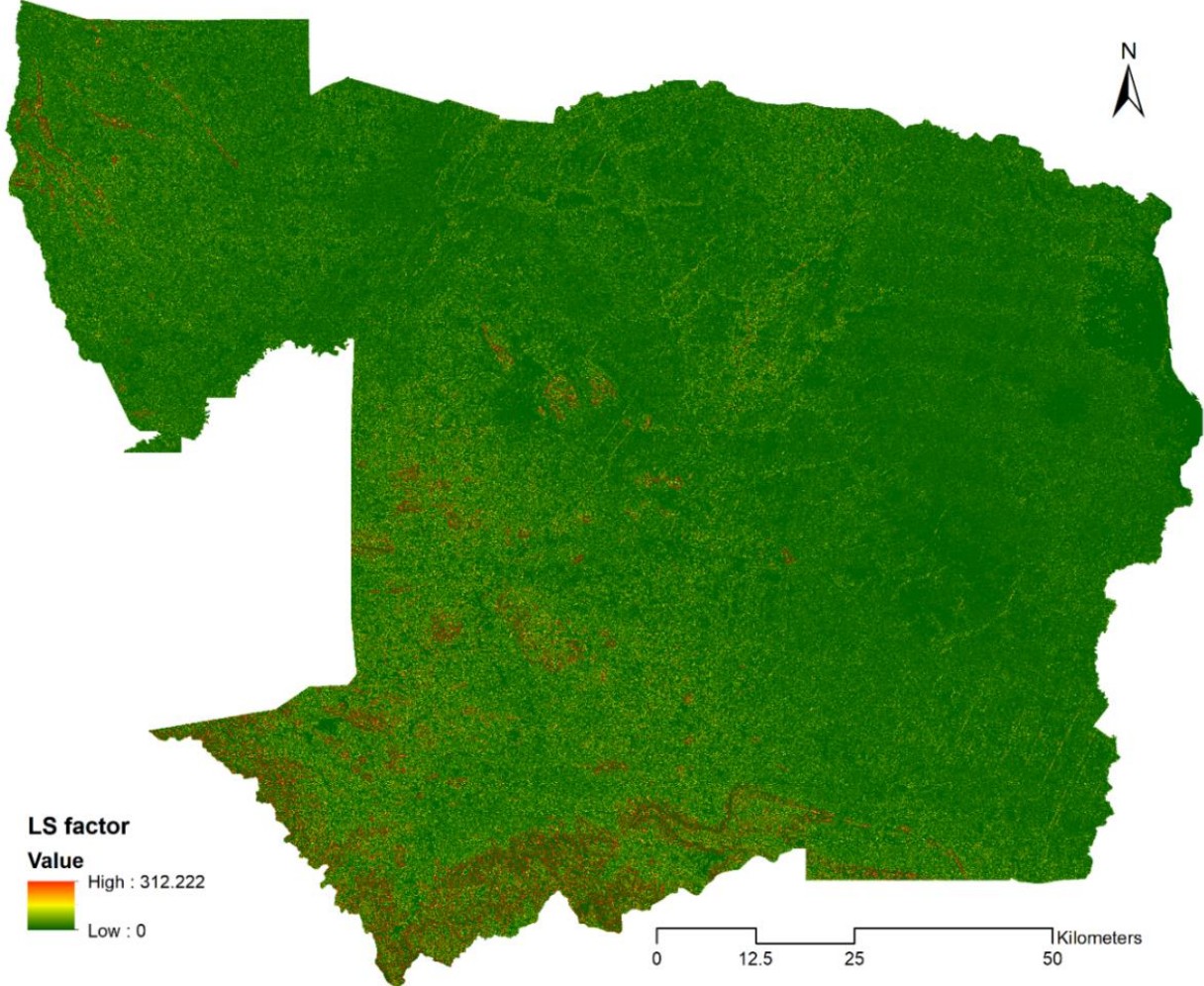

**Figure 6.** Spatial map of slope length and steepness (LS).

### 3.2.4. Cropping Management (C) Factor

The C factor was estimated by using the LULC map based on the literature review. The result of the C factors value ranged from 0.01 to 0.50 (Figure 7). The C average and the standard deviation were 0.24 and 0.23 in 1998, 0.32 and 0.22 in 2008, and 0.35 and 0.22 in 2018, respectively. In 1998, the predominant land of Battambang was forest cover in the west and southwest of the study area, especially Phnom Samkos wildlife sanctuary and Roneam Daunsam wildlife sanctuary, while the most forest cover in 2008 and 2018 almost finished at part of the western study area. The mean of C factor significantly increased

from 0.24 to 0.35 over the last two decades, in line with the decrease of forest cover from 30% to 6.18% (Table 4).

**Table 7.** Range of topographic factor (LS) and its area.

| Range of LS factor | Area | | Mean of LS Factor Range |
| --- | --- | --- | --- |
| | (ha) | % | |
| 0–5 | 1,182,154 | 98.22 | 0.34 |
| >5–15 | 17,408 | 1.45 | 8.24 |
| >15–25 | 2808 | 0.23 | 18.83 |
| >25–35 | 772 | 0.06 | 29.05 |
| >35–50 | 332 | 0.03 | 40.83 |
| >50–312.22 | 152 | 0.01 | 67.89 |
| Total | 1,203,628 | | |

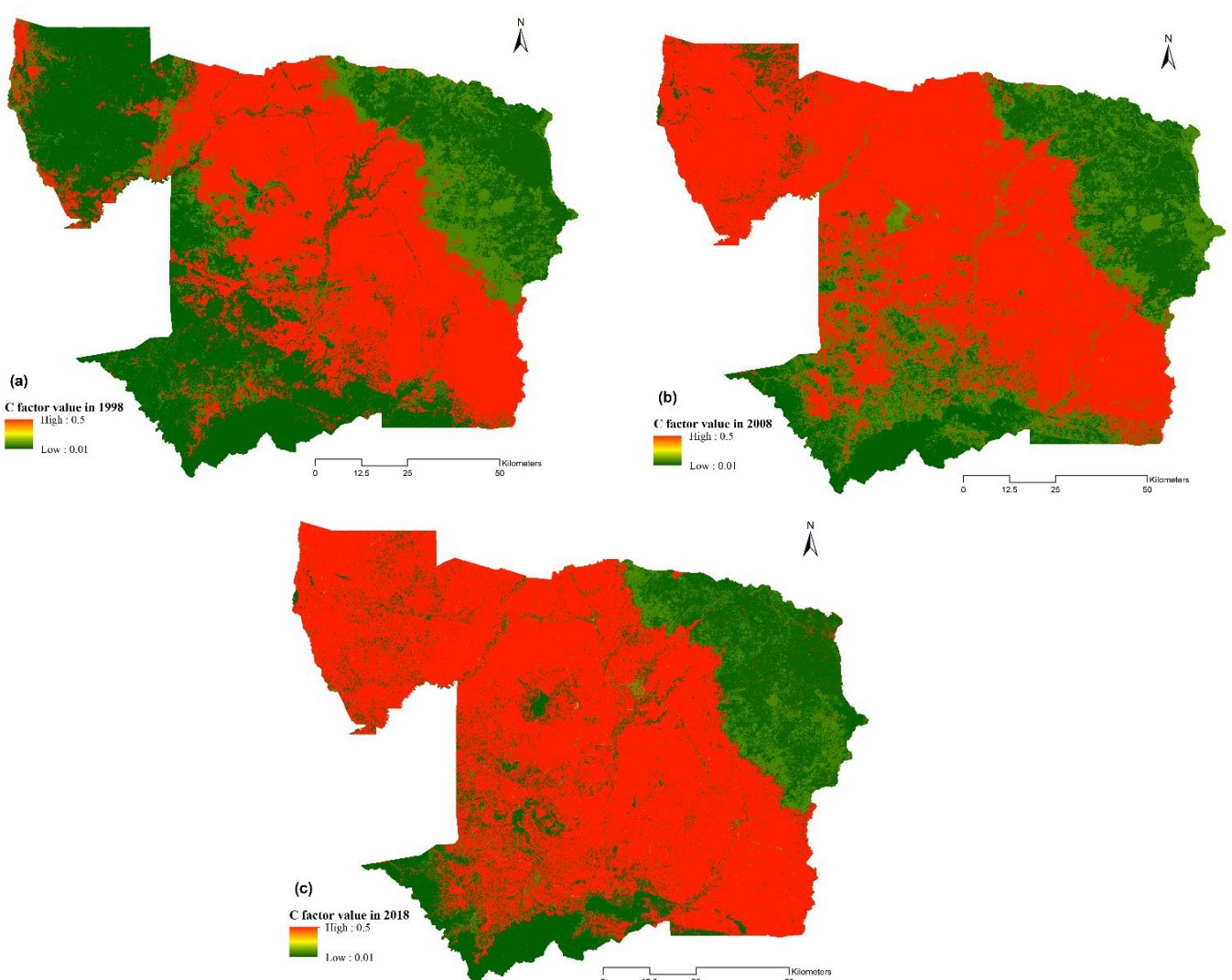

**Figure 7.** Spatial map of the crop management factor (C) of year (**a**) 1998, (**b**) 2008, and (**c**) 2018.

### 3.2.5. Supporting Practice (P) Factor

The supporting factor (P) varied from 0.1 to 1. The mean of P factor value in 1998, 2008, and 2018 was 0.61, 0.46, and 0.4, respectively, while its standard deviation was also 0.44, 0.43, and 0.41, respectively. For 2018, the largest area of a range of LS factor was 0 to 0.1, followed by the range of LS factor, 0.35 to 1. They were 37.44% and 31.96%, respectively.

In contrast to the largest area of the range of LS in 2018, the range of LS, 0.35–1, was the biggest in 1998 and 2008. They were 55.6% and 38.98%, respectively (Table 8 and Figure 8).

**Table 8.** Range of P factor responding to the area in 1998 to 2018.

| Range of P Factor | Area (2018) | | Area (2008) | | Area (1998) | |
|---|---|---|---|---|---|---|
| | (ha) | % | (ha) | % | (ha) | % |
| 0–0.1 | 450,690 | 37.44 | 424,341 | 35.26 | 329,770 | 27.4 |
| >0.1–0.12 | 221,805 | 18.43 | 199,088 | 16.54 | 136,906 | 11.4 |
| >0.12–0.14 | 117,693 | 9.78 | 94,173 | 7.82 | 59,909 | 5.0 |
| >0.14–0.19 | 18,487 | 1.54 | 11,407 | 0.95 | 6215 | 0.5 |
| >0.19–0.25 | 8081 | 0.67 | 4304 | 0.36 | 1535 | 0.1 |
| >0.25–0.35 | 2230 | 0.19 | 1346 | 0.11 | 336 | 0.0 |
| >0.35–1 | 384,641 | 31.96 | 468,970 | 38.96 | 668,957 | 55.6 |
| Total | 1,203,628 | | 1,203,628 | | 1,203,628 | |

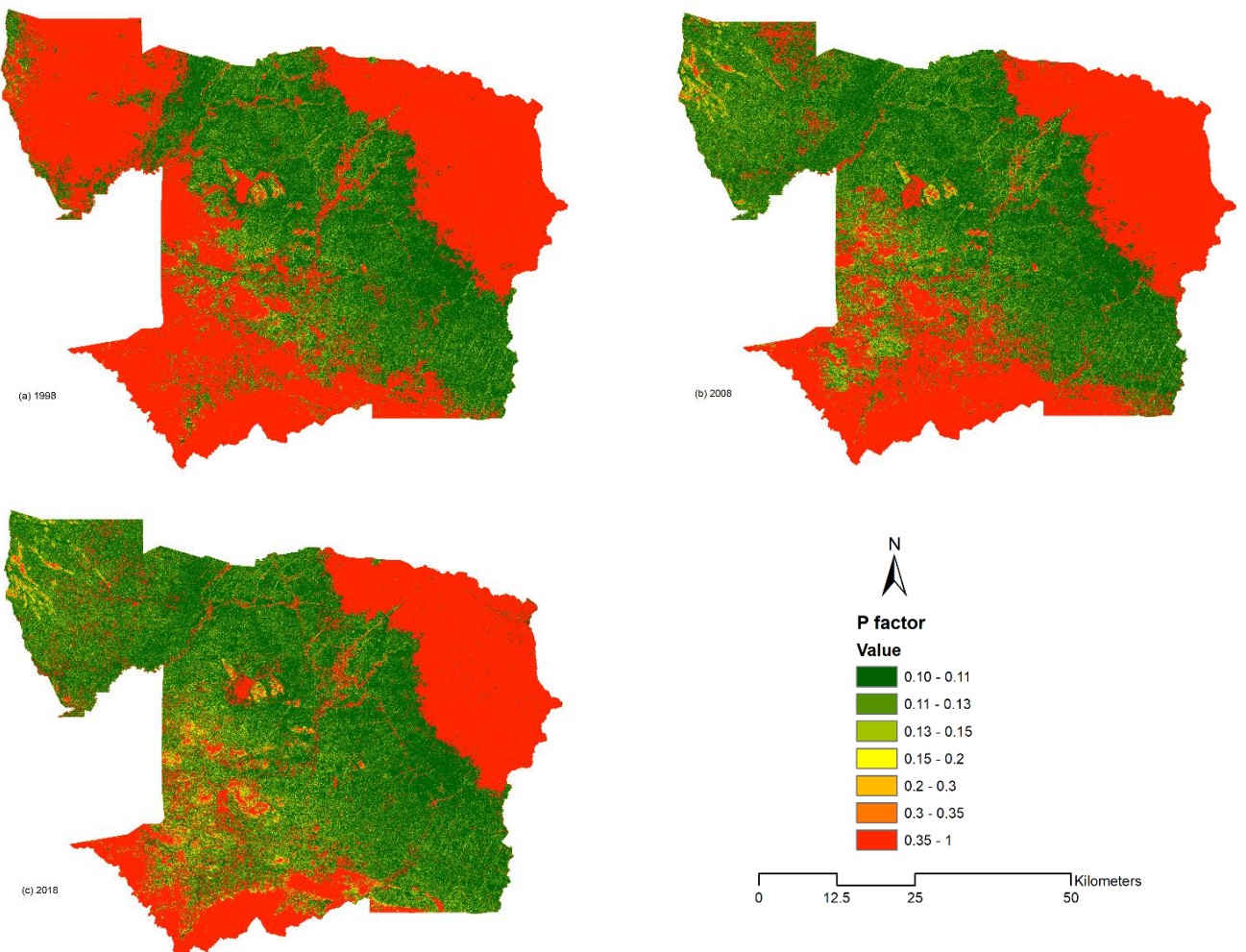

**Figure 8.** Spatial map of supporting practices factor (P) for year (**a**) 1998, (**b**) 2008, and (**c**) 2018.

### 3.2.6. Potential Soil Erosion and Actual Soil Erosion Estimation

The soil erosion was calculated by the RUSLE model in combination with the raster calculator of ArcGIS 10.3 over the last two decades. The actual and potential soil erosion was estimated. The supporting practice (P) and the cropping management factor (C) are significantly impacted on soil erosion [28]. Thus, only four factors, namely LS (L and S), R, and K factors, were used to estimate the potential soil erosion considered as a natural

erosion process without C and P factors [2]. The actual soil erosion was fully computed based on RUSLE variables. The potential soil erosion ranged from 0 to 17,090 t/ha.yr with an annual average and standard deviations were 68.39 t/ha.yr and 275.38 t/ha.yr, respectively (Figures 9 and 10). The mean of actual soil erosion experienced an increase of almost two times from 1998 to 2018 which was 2.92 t/ha.yr and 4.98 t/ha.yr, respectively (Figures 10 and 11). Most soil erosion occurred at the upland, particularly mountains, where forest cover decreased rapidly (Figure 11). The actual and potential soil erosion results were reported very differently due to the C and P factors, which were estimated by forest and cultivated areas. The results illustrated that C and P factors could reduce the rate of soil erosion from 2.65 to 4.52 times.

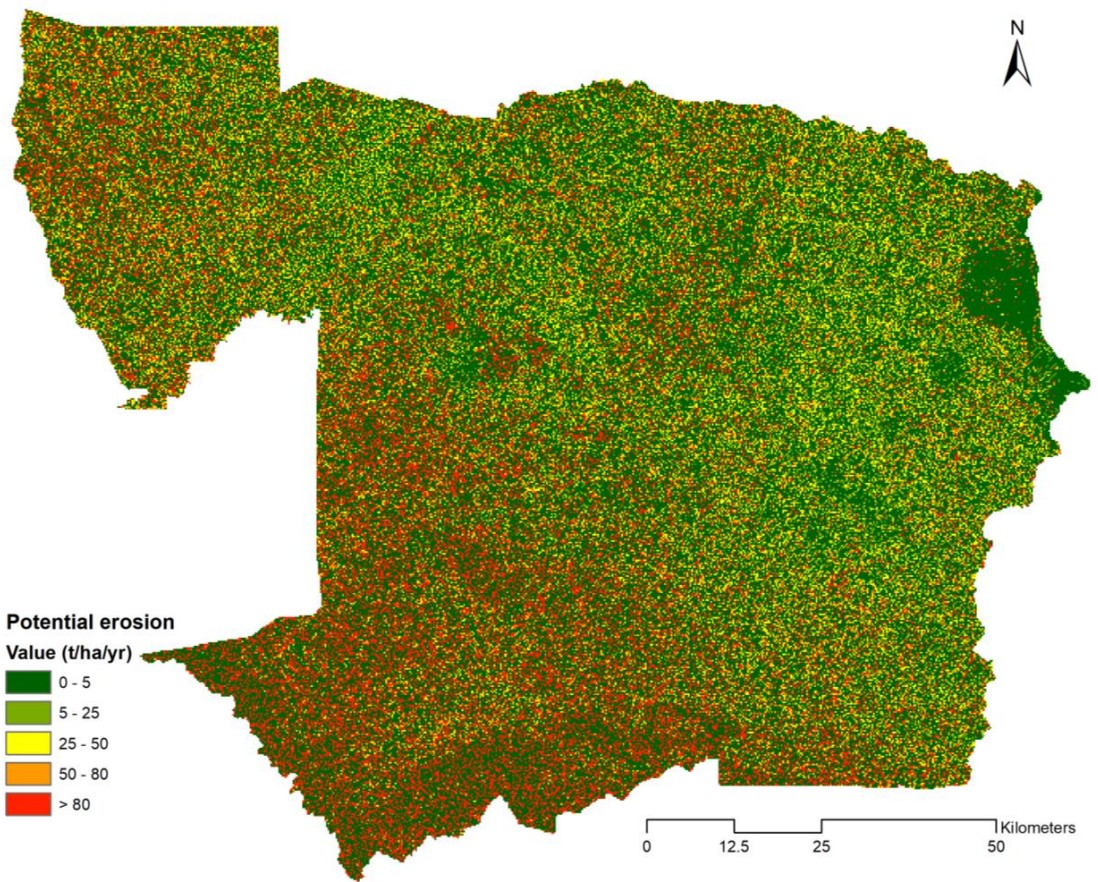

**Figure 9.** Spatial map of potential soil erosion.

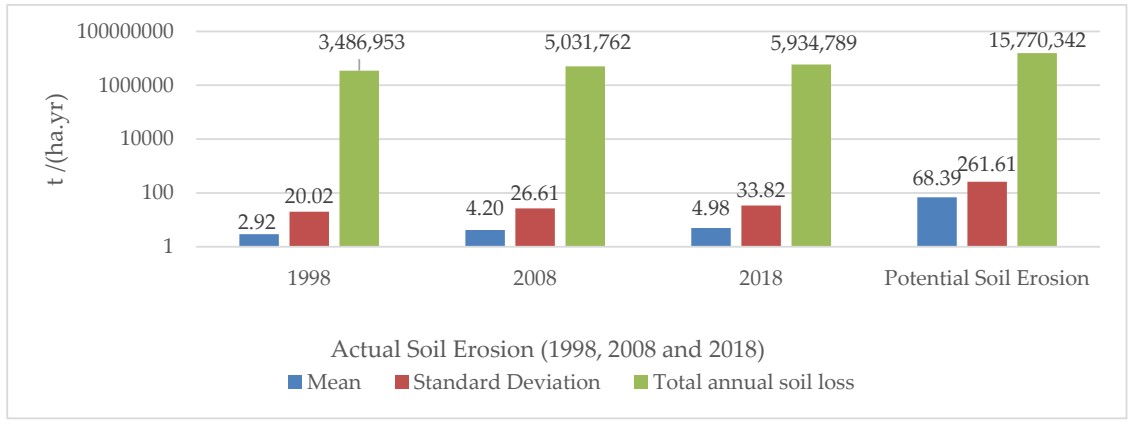

**Figure 10.** Mean, standard deviation, and total annual soil loss of actual and potential soil erosion.

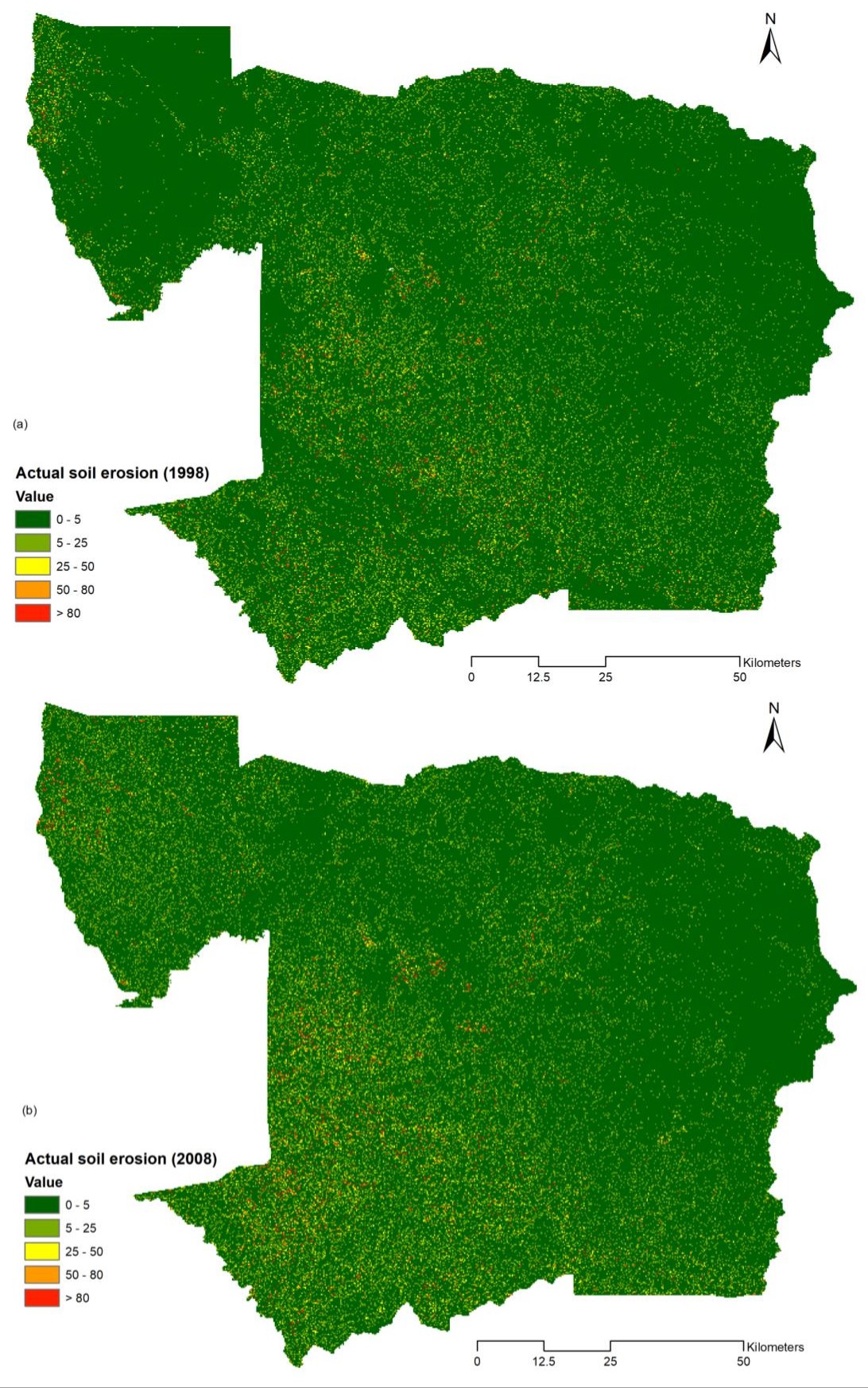

**Figure 11.** *Cont.*

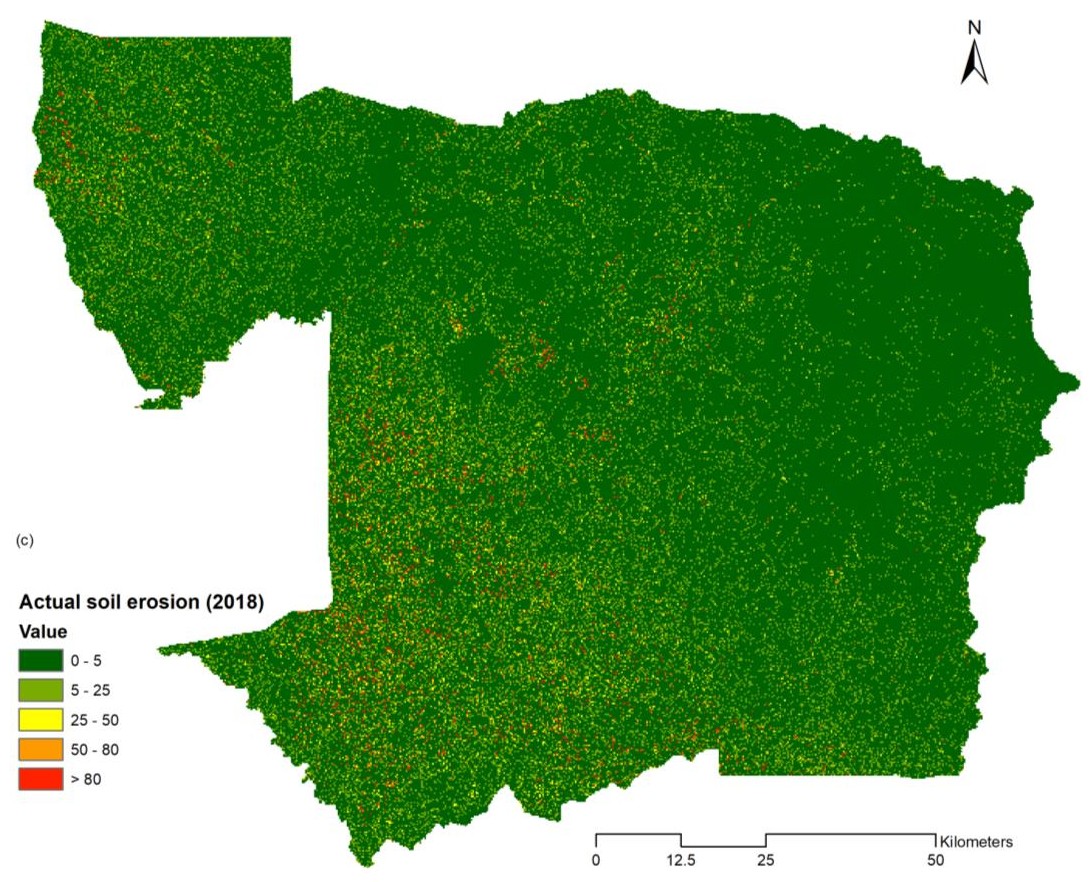

**Figure 11.** (**a**) Actual soil erosion in 1998, (**b**) actual soil erosion in 2008, and (**c**) actual soil erosion in 2018.

### 3.2.7. Soil Erosion Risk under Severity Classes

The revised RUSLE model in combination with ArcGIS 10.3 was used to estimate and to map soil erosion risk in accordance with reclassification and zonal statistics as the table of the spatial analyst tool, used to compute severity class of soil erosion rate over the last two decades (1998–2018) as given in Table 9. The actual soil erosion was categorized into five classes of severity such as: very low erosion (<5 t/ha.yr), low erosion (>5–25 t/ha.yr), fair erosion (>25–50 t/ha.yr), high erosion (>50–80 t/ha.yr), and very high erosion (>80 t/ha.yr) (Table 9 and Figure 8).

The predominant class in the study area was of very low erosion in 1998, 2008, and 2018. They were in 1,071,831 ha (89.05%), 1,014,429 ha (84.28%), and 1,012,021 ha (84.08%), respectively. In addition, the annual soil loss rate was 0.6 t/ha.yr in 1998, 0.61 t/ha.yr in 2008, and 0.58 t/ha.yr in 2018. In contrast to the largest area of severity class of soil erosion, the area cover of very high erosion (>80 t/ha.yr) of year 1998, 2008, and 2018 was 5803 ha (0.48%), 8881 ha (0.74%), and 11,945 ha (0.99%), respectively. The mean of annual soil loss rate was 2.92 t/ha.yr in 1998, 4.92 t/ha.yr in 2008, and 4.98 t/ha.yr in 2018. The fair severity class of 1998, 2008, and 2018 was 11,278 ha (0.94%), 17,539 ha (1.46%), and 16,454 ha (1.37%), respectively (Table 8; Figure 8).

### 3.2.8. Soil Erosion under LULC Classes

The spatial distribution of soil loss in the study area under LULC classes is shown in Table 10. The annual soil loss across agricultural land increased significantly from 1,909,348 (54.76%), 3,543,659 ton (70.43%), and 4,267,439 ton (71.91%) at the period of 1998–2018. Similarly, soil loss in 1998, 2008, and 2018 across shrubland experienced an increase of 244,976 ton (7.03%), 466,592 ton (9.27%), and 773,717 ton (13.04%), respectively. In contrast

to the rise of soil loss across LULC classes, forest cover and grassland decreased. Figure 12a also presents a local crop practice, leading to soil erosion.

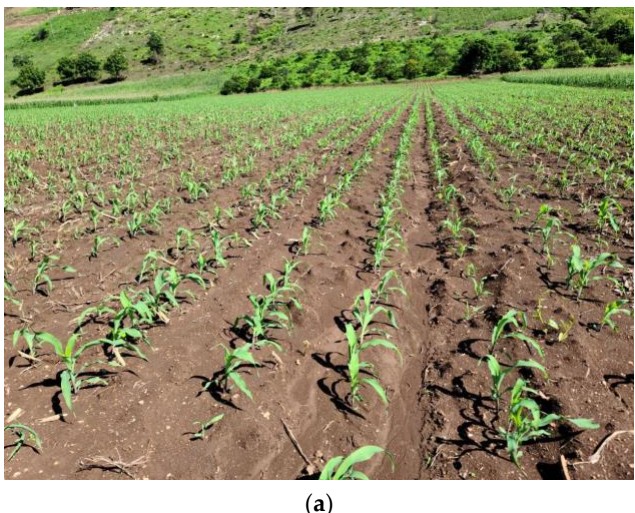

(**a**)

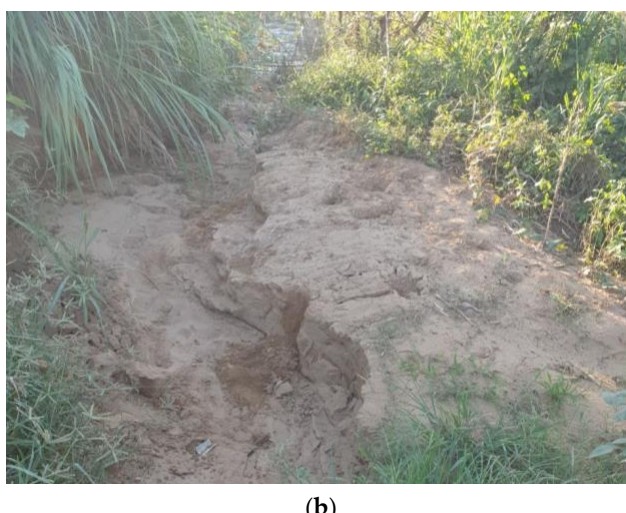

(**b**)

**Figure 12.** (**a**) Soil erosion under corn field in Borun village, Treng commune Ratanak Mundul District (17 September 2020) and (**b**) soil erosion under slope near Sangkae stream at Samlout district (8 October 2018).

3.2.9. Soil Erosion across Soil Types and Slope

There were a total 9 (nine) soil types, of which Acrisols, Cambisols, and Fluvisols were most predominant, with more than 94% of the study area (Table 11). Soil loss rate across different soil types was estimated as presented in Table 11. The soil erosion rates across almost all soil types changed, with the exception of the soil erosion rate across Arenosols. The soil erosion rates under Acrisols, Luvisols, and Ferralsols were found to be higher risk. The soil erosion rates across Acrisols (507,256 ha or 42.14%) were 4.15 t/ha.yr in 1998, 7.03 t/ha.yr. in 2008, and 8.71 t/ha.yr in 2018.

Topography is one of the major factors that impacts soil erosion. The slope of the study area was classified into 7 (seven) variety classes [28], namely; flat to very gentle slope, gentle slope, medium slope, strong slope, moderately steep, steep, and very steep, as illustrated in Table 12. The decrease of total annual soil loss across very gentle slope to medium slope was 52.28% to 42.38% over two decades. In contrast to the decrease, the total annual soil loss under moderately steep slope to very steep slope increased from 32% to 42% at the period of 1998–2018 (Table 12). Furthermore, Figure 12b showed the actual soil erosion under slope area along Sangkae stream in Samlout district, Battambang province.

**Table 9.** Soil erosion responding to severity classes in Battambang province.

| Severity Class | Soil Loss | LULC | | | | | | | | | | | | | | | Net Change | | |
|---|---|---|---|---|---|---|---|---|---|---|---|---|---|---|---|---|---|---|---|
| | | 1998 | | | | | 2008 | | | | | 2018 | | | | | 1998–2008 | 2008–2018 | 1998–2018 |
| | | Area | | Soil Loss | Total Annual | | Area | | Soil Loss | Total Annual | | Area | | Soil Loss | Total Annual | | | | |
| | (t/ha/y) | (ha) | % | (t/ha/y) | (ton) | % | (ha) | % | (t/ha/y) | (ton) | % | (ha) | % | (t/ha/y) | (ton) | % | (ha) | (ha) | (ha) |
| Very low erosion | ≤5 | 1,071,831 | 89.05 | 0.6 | 614,541 | 17.62 | 1,014,429 | 84.28 | 0.61 | 614,808 | 12.22 | 1,012,021 | 84.08 | 0.58 | 588,722 | 9.92 | −57402 | −2409 | −59810 |
| Low erosion | >5–25 | 107,282 | 8.91 | 10.4 | 1,112,077 | 31.89 | 150,318 | 12.49 | 10.67 | 1,604,603 | 31.89 | 143,992 | 11.96 | 10.70 | 1,540,322 | 25.95 | 43036 | −6326 | 36710 |
| Fair erosion | >25–50 | 11,278 | 0.94 | 34.5 | 388,651 | 11.15 | 17,53 | 1.46 | 34.19 | 599,683 | 11.92 | 16,454 | 1.37 | 34.18 | 562,426 | 9.48 | 6261 | −1085 | 5176 |
| High erosion | >50–80 | 5125 | 0.43 | 62.4 | 319,877 | 9.17 | 7752 | 0.64 | 62.43 | 484,009 | 9.62 | 8579 | 0.71 | 62.87 | 539,375 | 9.09 | 2627 | 827 | 3454 |
| Very high erosion | >80 | 5803 | 0.48 | 181.3 | 1,051,806 | 30.16 | 8881 | 0.74 | 194.64 | 1,728,660 | 34.35 | 11,945 | 0.99 | 226.37 | 2,703,944 | 45.56 | 3079 | 3063 | 6142 |
| Water | | 2309 | 0.19 | 0.0 | 0.00 | 0.00 | 4707 | 0.39 | 0.00 | 0.00 | 0.00 | 10,637 | 0.88 | 0.00 | 0.00 | 0.00 | | | |
| | | 1,203,628 | 100 | | 3,486,953 | 100 | 1,203,628 | 100 | | 5,031,762 | 100 | 1,203,628 | 100 | | 5,934,789 | 100 | | | |

**Table 10.** Distribution of soil loss across LULC classes.

| No. | LULC Classes | LULC | | | | | | | | | | | |
|---|---|---|---|---|---|---|---|---|---|---|---|---|---|
| | | 1998 | | | | 2008 | | | | 2018 | | | |
| | | Area | | Total Annual | | Area | | Total Annual | | Area | | Total Annual | |
| | | (ha) | % | (ton) | % | (ha) | % | (ton) | % | (ha) | % | (ton) | % |
| 1 | Built-up Areas | 48 | 0.00 | 127 | 0.00 | 302 | 0.03 | 57722 | 1.15 | 4698 | 0.39 | 35,100 | 0.59 |
| 2 | Water Features | 2309 | 0.19 | 0.00 | 0.00 | 4707 | 0.39 | 0.00 | 0.00 | 10,637 | 0.88 | 0.00 | 0.00 |
| 3 | Grasslands | 151,753 | 12.61 | 513,747 | 14.73 | 208,051 | 17.29 | 520,712 | 10.35 | 75,683 | 6.29 | 340,091 | 5.73 |
| 4 | Shrublands | 154,916 | 12.87 | 244,976 | 7.03 | 178,717 | 14.85 | 466,592 | 9.27 | 213,574 | 17.74 | 773,717 | 13.04 |
| 5 | Agricultural lands | 535,626 | 44.50 | 1,909,348 | 54.76 | 735,584 | 61.11 | 3,543,659 | 70.43 | 823,225 | 68.40 | 4,267,439 | 71.91 |
| 6 | Barren land | 16 | 0.00 | 54 | 0.00 | 224 | 0.02 | 13,068 | 0.26 | 1395 | 0.12 | 17,255 | 0.29 |
| 7 | Forest covers | 358,960 | 29.82 | 818,701 | 23.48 | 76,042 | 6.32 | 430,009 | 8.55 | 74,416 | 6.18 | 501,186 | 8.44 |
| | Total | 1,203,628 | 100 | 3,486,953 | 100 | 1,203,628 | 100 | 5,031,762 | 100 | 1,203,628 | 100 | 5,934,789 | 100 |

**Table 11.** Soil erosion across different soil types during 1998, 2008, and 2018.

| Soil type | Area | | 1998 | | | 2008 | | | 2018 | | |
|---|---|---|---|---|---|---|---|---|---|---|---|
| | | | Total Annual | | Soil Loss Rate (t/ha.yr) | Total Annual | | Soil Loss Rate (t/ha.yr) | Total Annual | | Soil Loss Rate (t/ha.yr) |
| | (ha) | % | (ton) | % | | (ton) | % | | (ton) | % | |
| Acrisols | 507,256 | 42.14 | 2,104,435 | 60.35 | 4.15 | 3,565,557 | 70.86 | 7.03 | 4,419,002 | 74.46 | 8.71 |
| Arenosols | 25 | 0.00 | 8 | 0.00 | 0.33 | 8 | 0.00 | 0.33 | 8 | 0.00 | 0.33 |
| Cambisols | 262,777 | 21.83 | 753,860 | 21.62 | 2.87 | 812,010 | 16.14 | 3.09 | 838,594 | 14.13 | 3.19 |
| Ferralsols | 6991 | 0.58 | 22,300 | 0.64 | 3.19 | 26,524 | 0.53 | 3.79 | 28,936 | 0.49 | 4.14 |
| Fluvisols | 368,939 | 30.65 | 499,288 | 14.32 | 1.35 | 522,376 | 10.38 | 1.42 | 527,537 | 8.89 | 1.43 |
| Gleysols | 42,550 | 3.54 | 70,205 | 2.01 | 1.65 | 65,429 | 1.30 | 1.54 | 70,875 | 1.19 | 1.67 |
| Lixisols | 319 | 0.03 | 229 | 0.01 | 0.72 | 842 | 0.02 | 2.64 | 871 | 0.01 | 2.73 |
| Luvisols | 6762 | 0.56 | 20,441 | 0.59 | 3.02 | 23,114 | 0.46 | 3.42 | 31,857 | 0.54 | 4.71 |
| Vertisols | 8008 | 0.67 | 16,186 | 0.46 | 2.02 | 15,902 | 0.32 | 1.99 | 17,109 | 0.29 | 2.14 |
| Total | 1,2036,28 | 100 | 3,486,953 | 100 | | 5,031,762 | 100 | | 5,934,789 | 100 | |

**Table 12.** Soil loss under slope severity classes.

| Slope Severity | Slope class (%) | Area | | 1998 | | 2008 | | 2018 | |
|---|---|---|---|---|---|---|---|---|---|
| | | | | Total Annual | | Total Annual | | Total Annual | |
| | | (ha) | (%) | (ton) | % | (ton) | % | (ton) | % |
| Flat to very gentle slope | 0–2 | 326,342 | 27.11 | 378,806 | 10.86 | 464,212 | 9.23 | 484,303 | 8.16 |
| Gentle slope | >2–5 | 314,096 | 26.10 | 600,649 | 17.23 | 752,902 | 14.96 | 815,747 | 13.75 |
| Medium Slope | >5–10 | 302,890 | 25.16 | 843,537 | 24.19 | 1,127,127 | 22.40 | 1,215,121 | 20.47 |
| Strong slope | >10–15 | 121,368 | 10.08 | 554,015 | 15.89 | 784006 | 15.58 | 927,776 | 15.63 |
| Moderately steep slope | >15–30 | 93,715 | 7.79 | 643,776 | 18.46 | 1,009,772 | 20.07 | 1,344,796 | 22.66 |
| Steep slope | >30–60 | 39,972 | 3.32 | 368,737 | 10.57 | 717,680 | 14.26 | 924,668 | 15.58 |
| Very steep slope | >60 | 5245 | 0.44 | 97,434 | 2.79 | 176,062 | 3.50 | 222,379 | 3.75 |
| Total | | 1,203,628 | 100 | 3,486,953 | 100 | 5,031,762 | 100 | 5,934,789 | 100 |

## 4. Discussion

The study used the Landsat 5 TM and Landsat 8 OLI images to produce LULC in 1998, 2008, and 2018. C factor and P factor were computed based on LULC and the combination between LULC and slope, respectively. In addition, soil data of the Soil Grids database of ISRIC-World Soil Information, DEM data, rainfall data, and satellite image were combined together to estimate soil loss in Battambang province. Overall, during the period from 1998 to 2018, the agricultural land experienced an increase of 54% equal to 287,600 ha (Table 4). This aligns with the estimation of World Bank (2015) which showed that a large share of past agricultural growth was driven by the expansion of cultivated areas from 2004 to 2012. On average, farmland increased annually by 4.7% over 2004–2012 due to deforestation [69,70]. They found that the forest cover declined dramatically to −79% (−284,500 ha) over 20 years. Similarly, Kong et al. [24] indicated that the forest cover of northwest of Battambang declined to 65% between 2006 and 2016. There was increase in built-up area due to population growth, infrastructure, and socioeconomic development [71–74]. The growth of migration of poor and landless farmers who accessed agricultural lands impacted LULC [24,75].

Soil erodibility, runoff rate, and soil exposure to erosion, as analyzed under the standard unit of plot condition, are defined as the K factor [76]. The highest K factor in our study was found at the upland areas and mountains, which is in agreement with others [2,77]. Overall, there were five different soil erosion severity classes (Table 9). This number of categories was based on Soil Erosion Standard Document–Technological Standard of Soil and Water Conservation (SD238–87) [78], and was used by the study [2]

for soil erosion prediction in the Mekong Lancang region, including Cambodia, as it does not have soil erosion classification. As in our study, the soil erosion severity classes were also reported in the Mekong Lancang region in 2019 [2]. In the previous research [2,23,79], RUSLE with GIS and remote sensing data were used to estimate soil loss.

Our results showed that the mean annual soil loss was 2.92 t/ha.yr in 1998, 4.20 t/ha.yr in 2008, and 4.98 t/ha.yr in 2018. Whereas the total annual soil loss was 3.49 million tons in 1998, 5.03 million tons in 2008, and 5.93 million tons in 2018. These results were in line with the finding of Nut et al. [25], which predicted that the soil loss would range from 3.1 t/ha.yr in 2002 and 7.6 t/ha.yr in 2015 in Stung Sangkae catchment of Cambodia. Chuenchum et al. [2] estimated the soil erosion rate in the Mekong Lancang region ranging from 7 to 10 t/ha.yr with an average of 5.35 t/ha.yr. Suif et al. [80] and Thuy and Lee [23] found that the annual mean of soil loss rate was 5 t/ha.yr in the Mekong Lancang. Similarly, losses were observed in the tropical conditions of Africa, for example, a study by Marondedze et al. [81] in Zimbabwe and Lufafa et al. [82] in Lake Victoria basin estimated an annual average soil loss of 5 t/ha/yr. Our results are also in accordance with previous studies made by [2,25,83], which showed that natural vegetation enables one to reduce and to protect against soil erosion, especially in cultivated land. Additionally, we analyzed the potential soil erosion (assuming C factor and P factor equal 1) and actual soil erosion (actual C factor and P factor) in order to assess the C factor and P factor's abilities to reduce and protect soil erosion (Figures 9 and 10). The results illustrated that the potential soil erosion was 2.65 to 4.25 times higher than the actual soil erosion due to the effect of the C and the P factor. This was similar to the finding of Chuenchum et al. [2], who reported that the C and P factors could reduce the rate of soil erosion from 2.5 to 7 times.

Furthermore, the results were also confirmed with a household survey in 2021 by the authors. The results showed that all respondents claimed that during 20 years from 1998 to 2018, soil fertility declined significantly. Figure 13 presented that 44%, 35%, and 7% of respondents reported that the soil fertility was in fair decline, strong decline, and very strong decline, respectively. However, according to the focus group discussion, the agricultural yield only slightly declined due to more use of chemical fertilizer than before. The chemical fertilizer consumption is increasing remarkably, and the local farmers spend more on chemical fertilizer. Similarly, other researchers confirmed that the soil loss made agricultural productivity decrease. The soil erosion could decrease corn productivity by 12% to 21% in Kentucky, 0–24% in Illinois, 25%–65% in Georgia, and 21% in Michigan, USA [84–86]. Additionally, Jie [87] reported that food production would decrease by 40% if the current rate of soil loss in China continues over the next 50 years.

The spatial distribution of soil loss under LULC classes varied in 1998, 2008, and 2018 (Table 10). The soil erosion rate change in the study area during 1998–2018 was correlated with an increase or decrease of erosion based on LULC classes. For instance, the annual soil loss across agricultural land increased significantly from 1,909,348 (54.76%) to 4,267,439 tons (71.91%) from 1998 to 2018 (Table 10). In contrast, the soil erosion under forest cover and grassland was the lowest. These results are in agreement with those of Chuenchum et al. [2,34] in the Mekong Lancang, Nut et al. [25] in Cambodia, Niacsu et al. [83] in Romania, and Gashaw et al. [88,89] in Ethiopia, who reported that the soil erosion rate broadly increased or decreased based on vulnerable LULC classes.

The agricultural land was the largest contributor to the total soil loss in Battambang province. Similar observations were made by other researchers; Nut et al. [25] in Stung Sangkae catchment of Cambodia, Fu et al. [90] in southeastern Washington State in USA, and Karamage et al. [91] in Rwanda, Africa, in which 81.5%, 92.8%, and 95% of total soil loss, respectively, was observed in agricultural land.

The study also found that forest cover, grassland, and shrubland were converted to agricultural land. They are also susceptible to soil erosion. Table 10 presented that the soil erosion under these areas was 45.24% in 1998, 28.17% in 2008, and 27.21% in 2018, respectively, while the soil erosion under cropping land was the highest, when compared to other layers. This finding also agreed with those of other researchers such as Nut et al. [25],

Pimentel and Kounang [92], and Patric [93], who claimed that the rate of erosion under agricultural land is more than that of these areas due to vegetation cover. If the agricultural land expansion is still occurring from the conversion of forest cover and grassland, and the poor agricultural practice still continues, the soil erosion will increase significantly. This would affect the farmers, who are facing the high expenses of chemical fertilizers in order to obtain higher agricultural yields. Hence, on-farm conservation agriculture practices (CAP), water conservation and management, agroforestry practices, vegetation cover restoration, and the creation of slope terraces should be applied to enhance sustainable land management, minimize damage to the environment, and function as adaption/mitigation measures against climate change [25,72]. Additionally, referring to consultation with local government, the local people understood the impact of soil erosion and CA project, especially the Appropriate Scale Mechanization Consortium (ASMC) project, which was implemented in Battambang province with the aim of focusing on scaling up conservation agricultural (CA) machineries through proper technologies and cropping systems and was actively pursued during Phase I [94].

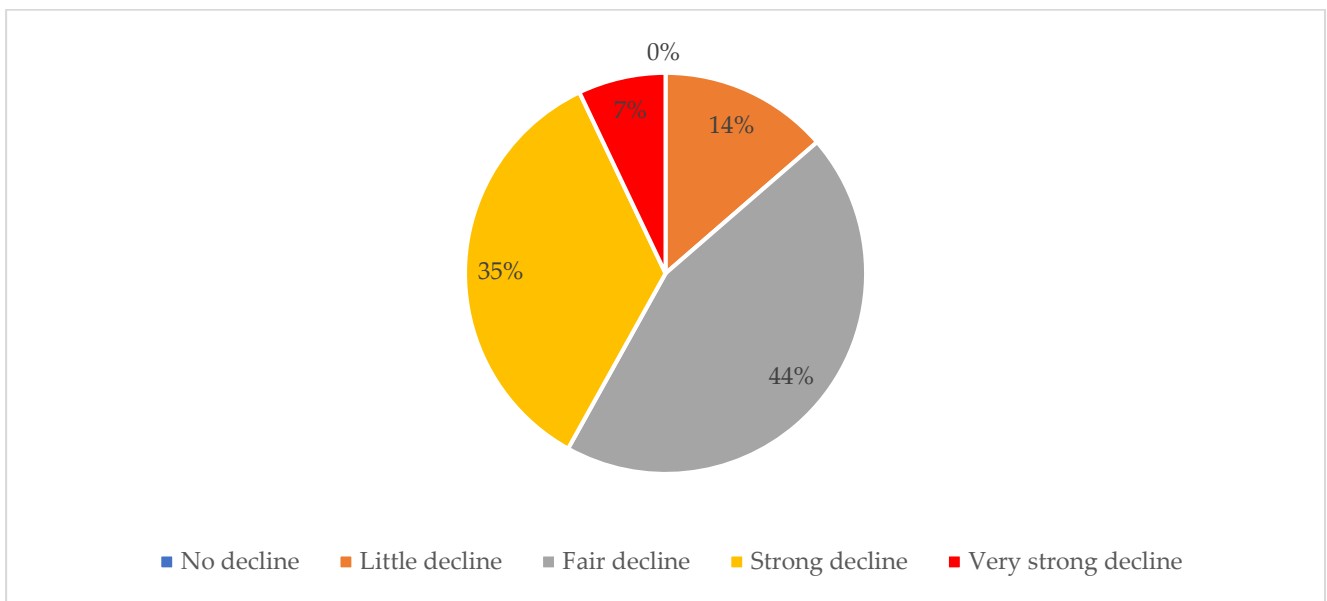

**Figure 13.** Perception of soil fertilizer decline during the last 20 years from 1998 to 2018.

## 5. Conclusions

Our results showed that the agricultural land, built-up area, water feature, shrubland, and barren land significantly increased, while the forest cover and grassland rapidly declined from 1998 to 2018 in the study region. As a result, the total annual soil loss increased from 3.48 to 5.93 million t/ha.yr between 1998 and 2018. The highest soil loss occurred in the uplands or mountains due to steep slopes and rapid deforestation. The agricultural land expansion significantly contributed to the total soil loss. Increasing soil erosion is a major challenge for the livelihood of people as it results in poor soil fertility and low crop productivity. Our research finding would be useful to agricultural experts or policy makers for developing strategies for reducing the rate of soil erosion in the high-risk zones (steep slope and low vegetation). There is an urgent need to mitigate and protect against soil erosion for sustainable, inclusive, resilient agriculture and water management. Measures to improve vegetation cover, use of conservation agriculture practices (CAP), along with water and soil conservation, management, agroforestry practices, vegetation cover restoration, the creation of slope terraces, and the use of direct sowing mulch-based cropping (DMC) systems should be considered. In addition to biophysical technologies, emphasis should be place on socioeconomic innovations, training producers, building social capital, and enhancing the capacity for agricultural extension.

**Author Contributions:** Conceptualization, T.S., S.P. and N.N.; methodology, T.S., S.P. and N.N.; software, T.S.; formal analysis, T.S., S.P. and N.N.; investigation, P.V.V.P., P.C. and D.T.; resources, T.S., N.N. and P.V.V.P.; data curation, T.S.; writing—original draft preparation, T.S., S.P. and N.N.; writing—review and editing, T.S., N.N., S.P., P.V.V.P., P.C. and D.T.; visualization, T.S. and N.N.; supervision, S.P., P.V.V.P., P.C. and D.T.; funding acquisition, P.V.V.P. All authors have read and agreed to the published version of the manuscript.

**Funding:** This research is made possible by the generous support of the American people provided to the Center of Excellence on Sustainable Agricultural Intensification and Nutrition (CE SAIN), to the Royal University of Agriculture (RUA), and to the project on "Pattern and Drivers of Land use Change in Battambang Province" implemented by the Faculty of Land Management and Land Administration, RUA, through the Geospatial and Farming Systems Research Consortium (GFSRC), University of California, Davis (United States) under bilateral agreement No. 201403286-06 between the two universities Subaward #S15115. Funds to CE SAIN and GFSRC were provided for research and scholarship through the Feed the Future Innovation Lab for Collaborative Research on Sustainable Intensification at Kansas State University, funded by the United States Agency for International Development (USAID) under Cooperative Agreement No. AID-OAA-L-14-00006. The contents are the sole responsibility of the authors and do not necessarily reflect the views of USAID or the United States Government or representing organizations.

**Institutional Review Board Statement:** Not applicable.

**Informed Consent Statement:** Not applicable.

**Data Availability Statement:** Data are available upon request from the corresponding author.

**Acknowledgments:** The author thanks Sanara Hor (Royal University of Agriculture, Cambodia), Robert J. Hijmans, Aniruddha Ghosh, and Alex Mandel (University of California, Davis) for their support and guidance in conducting this research. Contribution number 22-265-J from Kansas Agricultural Experiment Station.

**Conflicts of Interest:** The authors declare no conflict of interest. The funders had no role in the design of the study; in the collection, analyses, or interpretation of data; in the writing of the manuscript, or in the decision to publish the results.

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
