# Peer review of "Assessment of Land Use and Land Cover Changes on Soil Erosion Using Remote Sensing, GIS and RUSLE Model: A Case Study of Battambang Province, Cambodia"

_sustainability, doi:10.3390/su14074066_

Round 1

Reviewer 1 Report

The article Assessment of Land Use and Land Cover Changes on Soil Erosion Using Remote Sensing, GIS and RUSLE Model: A Case Study of Battambang Province, Cambodia is well written and well structured. The methodology and the data are correctly selected and performed. The methodlogy is well explanied. 

Some improvments need to be done, for example the R fator is being produced with small number of points, just 6. The interpolation for the R factor may benefit of more points or if no point are available, then use different interpolation method. If tryied, this should be stated in the manucript. 

The potential erosion map is not presented clearly. The selected numer of points for accuracy assessment seems not enough. The visual accuracy does not seem same with the statistical accuracy (over 80%). I recomend a test area and presented detaily. 

Author Response

Dear Reviewer:

            Thank you for giving us the opportunity to revise our manuscript. We appreciate comments and edits suggested by you (reviewer). You were certainly helpful to improve the quality of our manuscript. We are submitting revised version of our manuscript with track changes. We have highlighted the revised text in red colored font.  All the comments made by you (reviewer) were addressed and appropriate change were made in the manuscript. In addition, response to each of comments raised by you (reviewer) were submitted in the attachment.

            We sincerely hope that you (reviewer) are satisfied with our revisions and responses. Looking forward to your final decision.

Sincerely,

Authors

Reviewer 2 Report

Minor comments on the text and small technical errors:

  • in the abstract "The results showed that the area of agricultural land significantly increased from 23 44.50% in 1998 to 61.11% in 2008 and 68.40% in 2018" - it is not clear whether this data refers to the study area or e.g. the whole of Cambodia
  • Figure 1 Study area: - for which year is the current status? For 2022 or for 2018?
  • Figure 1 (b) - what does the term 'district' mean in the map legend; shouldn't it be 'district area' instead of 'District boundary'?
  • Figure 1 (b) what does 'Surveying locations' mean? What is this area? Is there any explanation for this in the text? Which surveys are involved here? I am also unclear about the icons labeled "District". Are they the centres of local administrative units or something else?
  • In Table 1. Slope severity in the study area based on FAO slope classification in column Slope class (%) shouldn't the values be separable, e.g. with the use of signs ≤ and > ? Similarly in other tables and figures.
  • Sangkae stream/Sangkae Stream - please standardize the spelling in the article, similarly with the term built-up area, because in the article the form: build-up area also occurred
  • There are shortcomings in the descriptions of figures and tables, where one could consider not only the description of the presented phenomenon but also on which area and in what period it occurs.
  • Figure 4 - there are no markings next to individual figures for the years 2008 and 2018.

General remarks:

- Based on the results obtained and the knowledge of economic and social processes in the agricultural areas of the study area, as well as the general socio-economic situation of the country, are the Authors able to make any prediction, even a very general one, of further changes in the described situation? For example, a comment can be added to Figs. 9 and 10. The latest data used in the study are from 2018.  Is the situation deteriorating further now, in 2021/2022, or are there any signs of improvement or slowing of soil erosion?

- Has it been determined in other studies or is it possible on the basis of the presented analysis to determine critical values for soil erosion processes for areas with similar natural characteristics and type and intensity of use? In other words, how much soil erosion are local ecosystems still able to withstand and where is the limit beyond which agricultural productivity will rapidly decline?

- The aims in the abstract raise the issue of sensitizing local actors to the scale of the soil erosion threat. Although this is not the main objective of the study, it would be good to see at least a small extension of the topics from the Conclusions section on how to reduce and prevent the described soil erosion processes. This would have a positive impact on the cognitive value of the article.

- Further, referring to the cognitive value as well as to the profile of Sustainability as an international, cross-disciplinary journal of environmental, cultural, economic, and social sustainability of human beings, it might be worth considering adding literally 2-4 photographs presenting an example of an area exposed to severe soil erosion (e.g. deforested, with a distinct slope), preferably in comparative terms in the studied period of 1998-2018.

- Remaining in the cross-disciplinary nature of the journal, a certain inaccuracy was noted by the reviewer in the passage on the social (qualitative/quantitative?) research methods used among the local population of the study area, which was not explicitly referred to later in the article, in the results, discussion, and conclusion sections.

- The article deals with extremely important and current problems from the point of view of ecology as well as a human activity. The authors presented well-documented and analyzed material and correctly described the methodology used. The selection of methods and research tools allows for a clear and sufficient description of the analyzed phenomenon. The research argument is clear and logical and the presented data are interrelated and form a coherent entity. Both earlier results of analyses from the area of research of other authors and references to works from other regions of the world have been cited. The methods of data presentation and the used figures reflect well the presented phenomena. Thus, the reviewed text is a solid and well-documented research material and meets all the requirements for scientific articles of a journal.

Author Response

(The authors gave the same response as above.)

Reviewer 3 Report

The assessment of land use and land cover changes on soil erosion using remote sensing and GIS with methodological RUSLE model approach of data processing, and mapping presented in this manuscript is important science-based contribution to the environmental and geo studies of Cambodia. The manuscript has all necessary components  and all sections are well-developed. Results of the manuscript have practical implications on environment affected by erosion in the region of Battambang Province, Cambodia and wider, in all other similar environment which are facing issue and consequences of erosion process. Manuscript fully and clearly describes goal, subject of matter, results and main findings of the research in a well-structured manner. All figures, tables and formulas are well presented and explained, and results are presented correctly, precisely and clearly. Conclusions are supported and well explained. All considered references are relevant and adequate, and all key relevant literature sources are cited. Authors have experiences in this field, and there was no gap in knowledge identified.         

            Authors explained why purpose of this study is important, what direct and indirect benefits (savings, increased yield..) this research may bring to environment, to government, to public, and how. However, authors may add more details about environmental problems, i.e. about severe land degradation and consequences. It is recommendable to add more details about environmental and agricultural problems, caused by erosion, for instance about extent of the damages in costs.

Row 59 – Laos instead of Loas

Row 61 – more? instead of less

Row 114 – Authors should shortly explain why they chose JICA classification, and not any other, as CORINE for instance. Also, JICA 2002 reference is missing, it have to be added in Reference list

262 – Did authors tried any other interpolation tool? Maybe Spline or Kriging would give more “realistic” results.

CORINE links which authors may consider:

https://land.copernicus.eu/user-corner/technical-library/corine-land-cover-nomenclature-guidelines/docs/pdf/CLC2018_Nomenclature_illustrated_guide_20190510.pdf

https://land.copernicus.eu/user-corner/technical-library/clc-product-user-manual

https://land.copernicus.eu/user-corner/technical-library/corine-land-cover-nomenclature-guidelines/html/index.html

Author Response

(The authors gave the same response as above.)

Round 2

Reviewer 1 Report

The authors did not completely revise the manuscript according to the comments but well explained why the revision was not possible.

At this point, the manuscript can be considered for publication.